# Microwave Modification of Quinoa Grains at Constant and Varying Water Content Modulates Changes in Structural and Physico-Chemical Properties of the Resulting Flours

**DOI:** 10.3390/foods12071421

**Published:** 2023-03-27

**Authors:** Ainhoa Vicente, Marina Villanueva, Pedro A. Caballero, José María Muñoz, Felicidad Ronda

**Affiliations:** 1Food Technology, Department of Agriculture and Forestry Engineering, University of Valladolid, 34004 Palencia, Spain; 2Department of Electricity and Electronics, University of Valladolid, 47011 Valladolid, Spain

**Keywords:** quinoa, heat moisture treatment, microwave treatment, techno-functional properties, thermal properties

## Abstract

Microwave-assisted heat moisture treatment (MWT) was applied to quinoa grains, a nutritious gluten-free pseudocereal of great interest in food product development, to achieve the physical modification of the quinoa flour. The effect of treating quinoa grains at different initial water contents (WC; 10%, 20%, and 30%) in two operational systems was compared: one open at atmospheric pressure and variable WC (V system), and the other in hermetic containers at constant WC (C system). The morphological structure of the obtained flours and their techno-functional, rheological, and thermal properties were evaluated. MWT proved to be effective in modifying these properties, the main effects probably being caused by protein denaturation and aggregation, and intragranular molecular rearrangements of starch, with disruption of short-range molecular order and even the partial collapse of starch granules in the samples treated at the highest WC. The greatest differences were observed for the 20 and 30% WC treated-samples, particularly when using C system, which increased their water absorption capacity and decreased their foaming, emulsion, and gel-forming capacities. Based on these results, the control of WC and its evolution during MWT of quinoa grains appears to be a viable and effective approach to adapt flour functionality to the needs of food production, allowing a wider range of flour properties depending on the MWT conditions.

## 1. Introduction

Quinoa (*Chenopodium quinoa* Willd.) is a pseudocereal of growing interest and popularity due to its excellent nutritional profile [1,2]. Quinoa is richer in protein than most cereals, containing approximately 15% protein with a balanced amino acid profile, being rich in lysine, a limiting amino acid in common cereals. In addition, it has a good oil fraction profile with unsaturated lipids, and it is a source of dietary fibres, polyphenols, minerals, phytosterols, and vitamins [1]. These nutritional properties offer great potential for functional food formulations. Another positive aspect is that it is a gluten-free (GF) grain, suitable for celiac and wheat-allergic patients [3]. 

In the last decade, quinoa has been used to make several food products such as bread, pasta, snacks, and GF bakery products [2,3]. In general, these foods presented a better nutritional profile but worse technological performance and sensory acceptability than those made with other ingredients such as wheat [2]. Quinoa, when used as a main ingredient in GF food products, cannot produce proper doughs as it has poor water and gas holding capacity, viscoelasticity, extensibility, ductility, and gel properties [4]. Therefore, it is necessary to modify the properties of quinoa flour to improve its technological and sensory performance in product development. 

One common approach to modify food functionality is through physical modification methods. These methods have been shown to be safe, non-toxic, and effective in modifying starch and flour properties, improving their range of application [5,6]. Among the physical methods, thermal processing is of great interest for starch-based ingredients, as it allows maintaining a “clean label” while achieving properties similar to those of chemically modified starches [7]. However, traditional heating methods, dominated by conduction heating, have low energy efficiency and do not meet the trend to reduce energy consumption and switch to “greener” technologies [4]. An interesting alternative to meet these new demands is the use of microwave radiation to perform heat-moisture treatments (HMT). In this regard, microwave energy is considered more efficient because it provides faster heating, more homogeneous treatment, and minimizes energy consumption [8]. 

Microwave-assisted hydrothermal treatment (MWT) has proved to be effective in modifying physicochemical properties and structure of starches and proteins. Microwave radiation produces the rearrangement of starch molecules, leading to a change in their morphology, crystallinity, functional, rheological, and thermal properties [9]. In terms of protein modifications, microwaves can change the internal structure of proteins and the degree of protein aggregation, affecting functional properties such as solubility, emulsification, stretching, and gelation [4]. In this regard, promising results were obtained to improve the textural properties of GF food gels by using microwave-heated quinoa protein [4]. Nevertheless, the literature employing this type of energy for heat treatment of more complex systems, such as flours and grains, is still limited although it has been proved that a greater treatment effect could be achieved by treating flour instead of starch, due to the presence of a high content of proteins, lipids, and non-starch polysaccharides [10]. 

When applying microwaves to perform HMT, the water content of the sample (WC) has proved to be critical in modulating the techno-functional properties of the treated flour and, therefore, its applicability, as demonstrated by studies applied to rice flour [8,11,12] and buckwheat grains [13]. These studies were carried out either by keeping the WC constant or by allowing the sample to dry during the treatment; however, the comparison between these two alternatives has not yet been studied. The analysis of the differences between these two options will provide valuable information on the effect of moisture and its evolution on the properties of flours, and will contribute to the development of ingredients with characteristics adapted to each specific food application. 

Traditional HMT treatments have recently been applied to quinoa. Almeida et al. [14] treated quinoa starch at 20% WC in a traditional oven at 110 °C for 1, 2, and 3 h, modifying its gelatinisation behaviour, relative crystallinity, and Fourier Transform Infrared Spectroscopy (FTIR) spectra, and obtaining agglomeration of starch granules. Dong et al. [15] treated quinoa flour at 30% WC in an oven at 110 °C for 90 min. They observed a reduction in the estimated glycaemic index, heat enthalpy, and relative crystallinity of the treated sample, as well as an increase in the interaction between starch and protein. Regarding MWT, quinoa grains have been microwaved for different purposes, such as roasting [16] and drying [17]. However, the study of physical modification of quinoa grains by short MWT to obtain quinoa flour with improved techno-functional properties and quinoa gels with modified viscoelasticity has not been performed so far. 

Therefore, the aim of this study was to explore the feasibility of MWT of quinoa grain performed under three initial WC (10%, 20%, and 30%), and two operation systems (one open at atmospheric pressure and variable WC, and the other in hermetic containers at constant WC) to modify the morphological structure and the techno-functional, thermal, and rheological properties of quinoa flour. This study will expand knowledge on the effect of critical control variables of MWTs and their ability to modulate the properties of treated quinoa flour to suit the needs of the food industry and allow its use in a wider range of products.

## 2. Materials and Methods

### 2.1. Samples

Quinoa grains (*Chenopodium quinoa* Willd.) cv. Titicaca were provided by Extremeña de Arroces (Cáceres, Spain). According to the supplier, saponins were previously removed from quinoa seeds by abrasion polishing. The proximal composition was (g/100 g quinoa): 15.6 ± 0.9% protein, 6.1 ± 0.6% fat, 2.4 ± 0.3% ash, and 11.0 ± 0.1% water content, measured with the 46-19.01, 30-10.01, 08-01.01, and 44-19.01 AACC official methods [18], respectively. Starch was 63 ± 1%, determined with the method for total starch described by Englyst et al. [19], and amylose content was 10 ± 1% of the starch, measured using the Concanavalin A method [20].

### 2.2. Microwave Treatment and Flour Obtention 

The WC of the quinoa grains was set to 10%, 20%, and 30%. Batches with 10% WC were obtained by drying the native grains in an incubation chamber (Memmert ICP260, Schwabach, Germany) at 35 °C. The 20 and 30% WC batches were obtained by adding distilled water and keeping the mixture in agitation using a rotary mixer MR-2L (Chopin, Tripette et Renaud, France) for 1 h, then storing it for 12 h at 4 °C to reach equilibration. 

Microwave treatments were performed in a customised microwave oven (R342INW, SHARP, Sakai, Japan) at 900 W and 2450 MHz following the procedures described by Villanueva et al. [11] and Vicente et al. [13]. A total of 100 ± 0.05 g of quinoa grains with the desired WC (10%, 20%, and 30%) were placed into two different Teflon^®^ containers: a non-hermetic container with small perforations in the lid that allowed variable WC during treatment as water could escape from the container (V system), and a hermetic container that maintained constant WC during treatment (C system). The treated samples were identified as V-10, V-20, V-30, C-10, C-20, and C-30, according to the type of container used and the initial WC of the grain. To allow a homogeneous distribution of the sample, a rotary system connected to a power supply unit was used to control the rotation speed of the container (70 rpm). For the microwave treatment, 48 cycles of 10 s exposure followed by 50 s of rest were performed (total microwave heating time of 8 min). Testoterm^®^ temperature strips (TESTO, Barcelona, Spain) placed inside the container in constant contact with the sample were used to measure the maximum temperature reached in each treatment. The final WC of each treated sample was also measured. According to their WC, the treated grains were conditioned to the initial/natural quinoa WC (11%) by moisturizing or drying them, with the methods described above. The conditioned quinoa grains from each of the different treatments, as well as native ones, were ground in a hammer mill (LM 3100, Perten Instruments, Stockholm, Sweden). Each microwave treatment was performed in duplicate.

### 2.3. Colour Characteristics

Flour colour was determined in sextuplicate with a PCE-CSM5 colorimeter controlled by the CQCS3 software. The CIELAB coordinates were used with a D65 standard illuminant and a 10° standard observer. L* (lightness from 0–black to 100–white) and chromatic coordinates a* (from green (–) to red (+)) and b* (from blue (–) to yellow (+)) were measured. The colour difference (ΔE) of each sample compared to the native flour was calculated using the following equation: ΔE = [(ΔL*)^2^ + (Δa*)^2^ + (Δb*)^2^]^1/2^. The Hue (h) and chroma (*C**) were also calculated from the CIELAB coordinates.

### 2.4. Particle Size Distribution

A laser diffraction particle size analyser (Mastersizer 2000, Malvern Instruments Ltd., Malvern, UK) was used to study the particle size distribution of the flours. The results were expressed as the diameter where the portion of particles with a smaller size is 10% (D_10_), 50% (D_50_ or median diameter), and 90% (D_90_). Samples were measured in triplicate.

### 2.5. Scanning Electron Microscopy (SEM)

A Quanta 200-F microscope (FEI, Graz, Austria) was used to study the morphological changes in the flours obtained after MWT and milling. The microscope was equipped with an X-ray detector, and samples were analysed in the low-vacuum mode, at an accelerating voltage of 5 keV, using a secondary electron detector and without prior metallization. Photomicrographs were taken at magnifications of ×3000 and ×12,000 to show the microstructure modifications.

### 2.6. Techno-Functional Properties

The water absorption capacity (WAC), oil absorption capacity (OAC), water absorption index (WAI), water solubility index (WSI), and swelling power (SP) of quinoa flour samples were measured at 5% concentration, as described by Abebe et al. [21] with modifications by Solaesa et al. [8]. WAC and OAC results were expressed as grams of water/oil retained per gram of flour dry matter (dm), WAI as g of sediment per g of flour dm, WSI as g of soluble solids per 100 g of flour dm, and SP as g of sediment per g of insoluble solids of flour dm. The foaming capacity (FC) and foaming stability (FS) of the samples were determined as described by Abebe et al. [21] at 2% concentration. Emulsifying activity (EA) and emulsion stability (ES) were determined as described by Vicente et al. [13]. The EA was expressed as a percentage of the volume of emulsion formed in relation to the initial volume. The ES was expressed as a percentage of the emulsion volume after heating in relation to the initial volume. Techno-functional properties were obtained at least in triplicate.

### 2.7. Pasting Properties

Pasting properties of the quinoa flour were determined with a Rapid Visco Analyser (RVA) model 4500 (Perten Instruments, Stockholm, Sweden) using the Standard 1 temperature profile of AACC official method 76-21.01 [18]. From the pasting curve, the pasting temperature (PT), peak viscosity (PV), trough viscosity (TV), breakdown viscosity (BV), final viscosity (FV), and setback viscosity (SV) were calculated. The determination was performed in duplicate.

### 2.8. Rheological Properties of Gels

The gels obtained as described in pasting properties measurement (Section 2.7) were used for dynamic oscillatory tests in a Kinexus Pro+ rheometer (Malvern Instruments, Malvern, UK) equipped with a parallel plate geometry (40 mm) with a serrated surface and a 1 mm working gap. The gels were placed between the plates and allowed to relax for 5 min at 25 °C. Strain sweeps were performed from 0.1 to 1000% strain at 1 Hz frequency. The linear viscoelastic region (LVR) was established, and the maximum stress (τ_max_) beyond which the dough structure broke and the stress at the crosspoint (G′ = G″) was determined. Frequency sweeps from 10 to 1 Hz were performed in the LVR at a constant strain of 0.5%. The frequency sweep data were fitted to the power-law model, as described by Ronda et al. [22]. The recorded viscoelastic parameters G′_1_, G″_1_, and (tan δ)_1_ are the coefficients obtained by fitting the frequency sweep data to the potential model, representing the elastic and viscous moduli and the loss tangent, respectively, at a frequency of 1 Hz. The a, b, and c are the exponents of the potential equation, and quantify the dependence of the dynamic moduli and loss tangent on the oscillation frequency. The complex modulus G^*^_1_ was calculated from (G′_1_+ G″_1_)^1/2^. All tests were performed in duplicate.

### 2.9. X-ray Diffraction (XRD)

XRD assay was performed using a Bruker-D8-Discover-A25 diffractometer (Bruker AXS, Rheinfelden, Germany) equipped with a CuKα radiation (λ = 0.154 nm) operating at 40 kV and 30 mA. Prior to measurement, all samples were equilibrated to 15% humidity using an incubation chamber (Memmert ICP260, Schwabach, Germany) at saturated humidity and 15 °C. Diffractograms and crystallinity were obtained as described by Villanueva et al. [11], in the range of 5–40° (2θ) at a rate of 1.2°/min, a scan step size of 0.02°, a divergence slit width of 1°, and a scatter slit width of 2.92°.

### 2.10. Fourier Transform Infrared Spectroscopy (FTIR)

FTIR spectra of flour samples (equilibrated at 15% WC) were recorded using a FT-IR Nicolet iS50 spectrophotometer (Thermo Fisher Scientific, Waltham, MA, USA) coupled to a crystal diamond attenuated total reflectance (ATR) sampling accessory. The analysis was performed in triplicate in the wavenumber range of 600–4000 cm^−1^ with a resolution of 4 cm^−1^ and accumulation of 64 scans. All spectra were normalized using OMNIC software (Thermo Fisher Scientific, USA). The short molecular order of starch was studied in the 1100 to 900 cm^−1^ region by estimating the changes in the relative intensities of the bands at 995, 1022, and 1045 cm^−1^ and determining the ratios 1047/1022 and 1022/995. Amide I bands (1700–1600 cm^−1^) were analysed using Origin 2019b (OriginLab Corporation, Northampton, MA, USA). The individual bands were determined on deconvolved curves (half-bandwidth 30 and enhancement factor 2.0) with a second derivative analysis followed by iterative fitting assuming Gaussian band shapes. The peaks found were assigned to a certain structure based on their frequency: β-turns (1700–1658 cm^−1^), α-helix (1658–1648 cm^−1^), random coil (1640–1648 cm^−1^), β-sheet (1640–1620 cm^−1^), aggregated strands (1620–1610 cm^−1^), and side chain vibration (1610–1600 cm^−1^) [23,24]. To estimate the percentage of secondary structural features, the percentage of area of the bands corresponding to each structure was used. 

### 2.11. Differential Scanning Calorimetry (DSC)

A differential scanning calorimeter (DSC3, STARe-System, Mettler-Toledo, Greifensee, Switzerland) was used to obtain the gelatinisation and retrogradation transition of the samples following the method described by Villanueva et al. [11]. The enthalpy (ΔH) values, expressed in J/g dm, and the peak (T_p_), onset (T_o_), and endset (T_e_) temperatures, expressed in °C, of gelatinisation and amylose–lipid peaks, were established. The enthalpy and peak temperature of the retrogradation were also recorded in 7 d stored samples at 4 °C to accelerate staling. Measurements were performed in duplicate.

### 2.12. Statical Analysis

Statgraphics Centurion 18 (Bitstream, Cambridge, MN, USA) was used for the statistical analysis. Analysis of variance (ANOVA) by applying least significant difference (LSD) test was used to assess the significant differences (*p* < 0.05) between samples. Differences were indicated in the tables by using different letters. Results were reported as the mean of different replicates depending on the parameter tested as indicated in previous sections and, for each parameter, the pooled standard error (SE) obtained from ANOVA was reported. A MANOVA study was carried out to evaluate the effect of the studied factors: the system used (V system and C system), the WC of grains during treatment (10%, 20%, and 30%), and their interaction. 

## 3. Results and Discussion

### 3.1. Colour Characteristics

Table 1 shows the colour parameters of the native and treated quinoa flours. Sample C-30 was the only one that showed colour differences appreciable by the human eye (ΔE > 5, according to Solaesa et al. [8]). This sample showed the lowest luminosity and hue, and the highest chroma; indicating a darker, reddish, and more vivid colour. The other treated samples also exhibited significant (*p* < 0.05) differences in some colour coordinates with respect to the control sample and with the same trend than C-30 but with much lower intensity. This difference may be mainly due to the thermal oxidation of polyphenols [25], which could represent a reduction in the antioxidant activity of the flour, and to the Maillard reaction, which could be enhanced by the formation of reducing sugars during MWT, that is favoured by a higher WC [8]. These processes would explain the higher colour differences for samples treated at high WC, particularly when it was maintained during the whole treatment, as happened when it was performed in hermetic containers.

As depicted in Appendix A, the colour difference was more pronounced on the outside of the grain than in the whole flour after milling. The thermal oxidation of polyphenols, present in higher concentration in the bran than in the endosperm [26], may be one of the causes of these observed differences.

### 3.2. Morphology and Particle Size Distribution

Figure 1 shows SEM micrographs of the endosperm particles of native and treated samples. Quinoa starch has very small polygonal granules (1–2 µm) that can occur in the endosperm individually and as spherically packed aggregates [1]. Image 1a shows a native quinoa flour endosperm particle with starch clusters within a matrix of individual components. Sample V-10 presented no appreciable structural differences with respect to the native flour. However, in C-10 at higher magnification (image 5b) a rougher surface appeared on the clusters. Similar effects have been observed after HMT of quinoa protein isolates [27] and MWT of rice flour [28] that were related to unfolded and/or denatured and aggregated protein bodies, spread over the granular surface. The largest differences were observed in sample C-30 (Figure 1, images 7a and 7b), that showed a completely different conformation to the native flour with the disappearance of the structure of compact aggregates and dispersed matrix, presenting larger irregular particles partially fused together. V-30 showed similar changes, but to a lesser extent, with some of the native structures still distinguishable. This reorganisation was probably caused by a partial gelatinisation of the outer layer of the starch granules due to the high moisture and temperature at the surface: protein and starch mixed and formed a new network of cross-linked protein and starch molecules [28,29]. V-20 and C-20 showed intermediate effects between those observed at lower and higher WC for each system. Dong et al. [15] found larger and more irregular particles due to particle aggregation, as well as a rougher surface with some cracks after treating quinoa flour with traditional HMT at 30% WC. Particle aggregation has been commonly reported after heat treatment of flours and starches [6,11,14,15]. However, no particle aggregation was observed in this study. This is probably caused by treating whole grains and not flours. Thus, after treatment, the moisture content of the grains was adjusted, and they were milled under the same conditions as the untreated grain.

Table 1 shows the particle size (D_10_, D_50_, and D_90_) measured with laser diffraction and Appendix A depicts the particle size distribution curves. In the fine fraction the differences were very slight, with no clear trend. However, in the coarse fraction not only were no agglomerations observed, but significant reductions in D_50_ and D_90_ were seen for all treated samples. This might indicate that physical modifications occurred during MWT in the grain components that reduced endosperm resistance against milling. 

### 3.3. Techno-Functional Properties

Table 2 shows the techno-functional properties of untreated and treated flour samples. Samples treated at 20 and 30% WC presented the greatest differences compared to the native flour samples, being more pronounced at higher and constant WC. These flours practically lost their ability to form emulsions and reduced their emulsion stability versus heating. Their foaming capacity and foam stability also decreased, dropping to zero for the latter property. These effects could be caused by the intense heating during MWT, which resulted in denaturation and aggregation of proteins [4]. These lager protein aggregates would have poorer interfacial properties to form stable foams and emulsions, as they would be unable to adsorb at air–water interface and efficiently cover up the fat droplets [27]. This loss of emulsifying and foaming properties must be taken into account when using treated flours for the production of products such as sauces or desserts. 

Samples treated at 20% and 30% WC exhibited higher WAC, with increases up to 114% for C-30. Solaesa et al. [8] previously reported higher water affinity after MWT of rice flour at WC above 15% and Liu et al. [30] after HMT of buckwheat starch, progressively increasing with higher WC. These authors related these effects to the increased hydrophilic tendency of starch molecules due to the disruption of hydrogen bonds between the amorphous and crystalline regions, with the expansion of the amorphous ones. In the case of C-30, the partial gelatinisation of starch, shown in DSC analysis (see Section 3.8), may also contribute to the higher water affinity. 

Treatments at 10% WC showed no effect in WAC and EA, but ES was slightly reduced. C-10 had a small increase in FC compared to native flour, but FS decreased. For these samples, the lower water availability and the lower temperature reached in MWT may have reduced the effects on protein denaturation and starch rearrangement, resulting in mild or non-significant changes. 

OAC presented little variation, with a slight but significant (*p* < 0.05) reduction in the V-20, V-30, and C-30 samples. A decrease in the hydrophobic tendency, shown by a lower OAC, was also observed for roasted quinoa grains [16]. These authors related it to a reduction in molecular interactions between starch molecules, and between starch and lipids, as well as the increased hydrophilic nature of molecules as shown by the higher WAC values. 

WAI and SP varied significantly (*p* < 0.05) with MWT, depending on the WC and the treatment system of the grains. The double interaction did not affect these properties (Table 2). WAI and SP slightly increased for treatments at 10% WC, particularly with the C-system (C-10) (+6% for both), but decreased for 20% and 30% WC samples, and more markedly with the V-system. WSI is related to the amount of soluble solids and is often an indicator of starch molecule degradation and dextrinization [31]. This parameter showed no clear trend, with increases for C-30 and decreases for V-20, C-10, and C-20. After HMT of starch, a decrease in SP and solubility was generally found; however, depending on the matrix and treatment conditions, increases in these parameters have also been reported [32]. Almeida et al. [14] reported increases in SP and solubility of quinoa starch treated at 20% WC and 110°C for 1, 2, and 3 h under conventional heating. However, when MWT was performed on buckwheat grain, Vicente et al. [13] reported increases in WAI and SP at lower WC, and reductions at higher WC, results that were similar to those observed in this study. The effect of MWT at different WC and evolution on the internal rearrangements of the starch granules and the protein denaturation could have influenced the differences observed. 

### 3.4. Pasting Properties

Pasting profiles of native and quinoa-treated flours are plotted in Figure 2 and the pasting parameters are given in Table 3. 

Results showed a progressive rise in the PT with flour WC, with the effect being more intense in the C-system (up to 7 °C for the C-30 sample). A greater resistance to swelling and rupture of starch granules has also been observed for MWT and HMT of starch from various origins, and was associated with increased intragranular molecular rearrangement [9,32]. PV and TV showed increments for V-10 and V-20, but decreased in the remaining samples, particularly in C-30, which led to a nearly flat pasting profile. Higher viscometric profiles, as obtained in V-10 and V-20 samples, have been previously reported for dry heat treatment of quinoa starch [6]. However, it differs from the usual behaviour reported for starch treated by HMT, in which the peak viscosity normally decreases gradually as the moisture level increases [33]. Collar [34] reported an enhanced viscosity profile for HMT of barley, buckwheat, and wheat flour at 15% WC and 120 °C for 1 h. This author related this behaviour with an increased hydrophobicity of prolamins and glutelins when the treatment was applied at low WC and short heating time to high-protein flours. However, the lower PV of samples treated at the highest WC, 30% WC, particularly when it was maintained constant during the whole treatment (as happened when was used the C-system), could be related to a partial gelatinisation. Starch disintegration could change the surface properties of granules and decrease their viscosity parameters as a consequence of an increase in hydrophilic bonds and the disruption of amylopectin molecules into smaller fractions [29]. 

BV was reduced for all treated samples, especially at higher WC, indicating higher stability of starch versus heating and shearing [33]. The SV value showed an increase with flour WC until it reached a maximum at 20%, and then decreased in the samples treated at 30%. The effects were more pronounced in the samples treated with the C-system. Thus, sample C-20 presented the highest SV value, 82% higher than the control, and C-30 the lowest one, 45% lower than the control. Except for sample C-30, all treated samples increased their SV values with respect to the untreated sample, meaning an enhanced amylose retrogradation ability. This effect is opposite to that usual found in MWT and traditional HMT of starches, where SV was reduced [9,33]. This fact, similarly to the case of PV, could be related to the effect of the high protein content of quinoa [34]. Moreover, the joint effect of having a high amount of protein and the lower temperature reached by these samples (below 100 °C, when most of the HMT studies were performed at 100 °C or higher) might have modulated the intragranular molecular rearrangements of starch leading to a higher amylose reassociation. 

### 3.5. Rheological Properties of Gels

The rheological properties of the gel samples were determined by dynamic oscillatory tests. Table 3 presents the parameters obtained from fitting the frequency sweep data to the power law model and the maximum stress (τ_max_) within the LVR and the crosspoint (G′ = G′′ and tan δ = 1) from the strain sweep. The frequency sweep and the strain sweep plots are presented in Figure 3.

The value of τ_max_ decreased for V-20, V-30, and C-30, indicating a weaker structure that is less resistant to stress breakage, especially for C-30 with a 92% reduction. The stress at which the gels changed from solid-like to viscous-like behaviour (crosspoint) increased slightly for 10% WC treatments and decreased markedly for the 30% WC ones. Therefore, the samples treated at 10% WC required a higher effort to collapse their structure after leaving the LVR, even though they did not show a higher resistance to stress. Vicente et al. [13] also reported a weaker structure for gels made from microwaved buckwheat grain at 30% WC and related it to a partial gelatinisation during MWT.

Frequency sweeps showed that all viscoelastic properties were significantly affected by the studied factors (WC and system) and their interaction, except for the exponent “a”. G′_1_ decreased with MWT, especially at 30% WC, while the exponent “a” increased for 20 and 30% WC treatments, showing a higher frequency dependence of this parameter, but still low. G′′_1_ slightly increased for V-20 and C-20, but decreased for the other samples, except for V-10, which remained unaltered. The exponent “b” increased in the samples where G′′_1_ decreased, showing a stronger frequency dependence of the less viscous samples. These variations only led to increases in the loss tangent of V-20, C-20, and V-30 samples, denoting a reinforcement of its viscous behaviour with regard to the native gel. The other samples did not differ from the native one as the G′′_1_/G′_1_ was kept constant in spite of the variation of the individual values of the viscoelastic moduli. After MWT of rice flour [8] and buckwheat grains (except for 30% WC) [13], more stable and consistent gels have been reported, with a reinforcement of the elastic behaviour. The lower temperature reached by the quinoa samples, while in the other studies it was higher than 100 °C, as well as the different structures of quinoa with high protein content and very small starch granules with low amylose content (10%), may have caused these different results.

### 3.6. X-ray Diffraction (XRD)

Figure 4 shows the XRD pattern of the quinoa samples. Quinoa flour presented an A-type diffraction pattern with peaks at 15.3°, 17°, 18°, 20°, and 23.4°, characteristic of cereal starches, and a crystallinity of 42.6%, which is in agreement with previous studies on quinoa [1]. 

MWT did not change the starch crystalline pattern type, thus showing a similar arrangement to that of the native flour and suggesting that the crystal structure was not highly modified by MWT. However, for some of the samples, the intensity of the peaks varied, with C-20 and C-30 being the most changed samples, but in opposite ways. Granular starch semicrystalline structure can be affected differently depending on processing parameters, botanical structure origin, and moisture content. Both increases and decreases in crystallinity after MWT have been reported [9,11,35]. The higher peak intensity and crystallinity showed by C-20 (+2.6% with respect to the control flour) could be related to the formation of a more ordered crystalline matrix caused by the displacement of the double helical chains within the starch crystals [36]. However, in the case of C-30, a different process occurred with a reduction in the relative intensity of the main peaks and of the crystallinity (−3.4%). The higher temperature and water availability in C-30 could have caused the disruption of the starch structure, leading to a reduction in crystallinity. This effect was confirmed by the reduced gelatinisation enthalpy shown in DSC (see Section 3.8) and the lower pasting profile (see Figure 2). 

### 3.7. Fourier Transform Infrared Spectroscopy (FTIR)

Figure 5A presents the FTIR spectra of the quinoa samples in the wavenumber range 4000–400 cm^−1^. FTIR spectra were studied in the range of 1100–900 cm^−1^ (region for short-range/double helical order of starch) for starch analysis, and in the range of 1700–1600 cm^−1^ (amide I band region) for protein secondary structure analysis (Table 4). 

The amide I band, whose vibrations are mainly due to C=O stretching vibration of the amide groups (approximately 80%) and some in-plane N-H bending (<20%), is suitable for studying changes in protein secondary structure due to its high sensitivity to small variations in molecular geometry and hydrogen bonding patterns [37]. The deconvoluted spectra is presented in Figure 5B and their fit to gaussian band shapes are depicted in Appendix A. The relative area assigned to each structure is shown in Table 4. Protein secondary structure changed significantly (*p* < 0.05) with the WC of quinoa grains during the treatment for all conformation types. However, some structures were not affected by the treatment system (side chain vibration, aggregated strands, and random coil) or by the WC*system interaction (random coil and α-helix). The aggregated strands increased with MWT (except for V-10). Increases in β-sheet and reduction in β-turns were observed in variable moisture system and C-10 samples (which correspond to the softer treatments, with lower initial and/or final WC and lower temperatures). The enhancement of β-sheet accompanied by the exposure of hydrophobic residues could promote the formation of protein aggregates [4]. Besides, the increased aggregated strands may be attributed to protein aggregation after polypeptide unfolding, at the expense of β-turn structure [24]. Therefore, these modifications in the secondary structure of proteins are compatible with protein aggregation and are in agreement with the reduced emulsifying and foaming capacities observed in the techno-functional properties. The random coil was reduced in all samples except C-20 (for this sample, the reduction was found in the α-helix). The reduction in the random coil denoted a structuring effect of MWT, as the creation of new regular hydrogen bonding pattern within the polypeptides led to a more ordered β-sheet-like structure. Opposite results were observed by Kheto et al. [16] when roasting quinoa without moisturizing at 300 W, 600 W, and 900 W for 5, 10, and 15 min, obtaining increases in random coil structure and reductions in β-sheet. These opposite results demonstrate the potential of microwave treatment to modulate the outcomes obtained as a function of treatment conditions. Besides the changes in proteins secondary structures, a shift towards higher wavenumber was observed in the position of the peaks of the deconvoluted spectra of MWT flours (Figure 5B), with this shift being more remarkable for samples V-20, V-30, and C-20 (up to 6 cm^−1^). Changes in the wavenumber position of some of the peaks in the deconvoluted spectra were reported previously for roasted chickpea [38]. These authors indicated that this effect confirmed that changes in protein conformation had occurred, even within the same structure type. 

Concerning the evaluation of short molecular starch order, the ratios 1047/1022 and 1022/995 were studied. The band at 1045 cm^−1^ is related to the crystalline structure of starch and the one at 1022 cm^−1^ to the amorphous structure. The absorbance band at 995 cm^−1^ is linked to water–starch interactions, resulting from the bonding in hydrated carbohydrate helices (due to C-OH bending vibrations). The 1047/1022 ratio has been used to quantify the degree of short-range order in starch and the 1022/995 ratio to obtain information on the state of organization of the double helices located within the crystallites [37,39]. Native quinoa flour presented 1047/1022 and 1022/995 ratios of 0.80 ± 0.02 and 0.94 ± 0.01 respectively, similar to those reported by Velásquez-Barreto et al. [39] for different quinoa varieties, whose values were 0.67–0.77 and 0.92–0.97 respectively. The 1022/995 ratio showed practically no variation with MWT, indicating that MWT had little or no effect on the hydrated to disordered structure proportion. However, the 1047/1022 ratio significantly decreased with MWT up to 6%, indicating disruption of short-range molecular order, related to variations in the ratio of the amounts of ordered to unordered fraction within the starch granules. Reductions in 1047/1022 ratio were previously reported by Li et al. [40] for microwaved millet starch and by Almeida et al. [14] for HMT quinoa starch, indicating that the crystal regions were less ordered in treated samples.

### 3.8. Differential Scanning Calorimetry (DSC)

Thermal properties determined from gelatinisation (first) and retrogradation (second) scans are given in Table 5. In the gelatinisation scan, two peaks were observed, as previously reported for quinoa flour [1]. The first peak, at lower temperature, was related to starch gelatinisation (melting of amylopectin crystallites), while the second, at higher temperature, was assigned to the disruption of the type I V-type crystallites of amylose–lipid complex [41]. Gelatinisation associated enthalpy, ΔH_gel_, only varied significantly with respect to the native sample for V-10 (+7%) and C-30 (−31%) samples. The enthalpy increase for V-10 could be caused by the recrystallization of starch during MWT at low WC (this sample started the MWT at 10% WC and ended it at 8% WC), making the starch structure more stable and thus requiring more energy to destroy the crystalline area [6]. However, the reduction in C-30 may be due to a partial gelatinisation of the starch granule allowed by the higher moisture availability in this treatment (30% WC during the whole treatment) and the highest temperature reached, as Solaesa et al. [8] reported for MWT of rice flour. Samples treated at 10 and 20% WC had a lower T_p-gel_ than the untreated sample (around −1 °C) while the sample C-30 presented a higher T_p-gel_ (+0.7 °C). However, only C-30 showed a significant delay in T_o-gel_ and T_e-gel_ compared to the native sample, with increases of 1.3 °C and 1.4 °C, respectively. These small temperature changes could be caused by the structural reconfiguration in the starch macromolecule during MWT, forming crystallites of different stabilities, as well as the formation of smaller chains and the change in amylose/amylopectin ratio [9]. The amylose–lipid complex presented a lower enthalpy, ΔH_am-lip_, for samples treated at 20% and 30% WC, while the peak temperature, T_p-am-lip_, was lower for all treated samples. Different effects on the amylose–lipid complex after MWT have been previously reported: ΔH_am-lip_ was reduced for rice flour when treated at 30% of initial WC in open containers that allowed drying of the sample during treatment (V system) [11], but no differences were observed in rice flour treated at constant WC [28] (C system) or in buckwheat flour from MW treated-buckwheat grains [13] (C system). Different matrix and treatment conditions, such as temperature, initial WC, and treatment conditions (V or C system) may have greatly affected the formation of the amylose–lipid complex.

The retrogradation scan, applied to gelatinised samples stored for 7 d at 4 °C in the DSC pans, also resulted in two peaks. The first one, related to the melting of recrystallised amylopectin, was very broad and had a low peak temperature. The retrogradation peak always presented a low enthalpy, ΔH_ret_, despite the applied storage conditions, long period, and low temperature that accelerate the amylopectin recrystallization [42]. No statistically significant differences for all treated samples with respect to the native one (except for C-30) were obtained, denoting a limited effect of MWT on the amylopectin ability to reassociate after gelatinisation in samples treated under mild treatment conditions [8]. However, the sample treated at the highest WC, 30%, for the whole treatment period (C-30), which reached the highest temperature (around 100 °C), increased its retrogradation tendency, as was reported previously for MWT rice flour [11]. The second, and reversible, peak corresponding to amylose–lipid dissociation had a higher enthalpy, ΔH_am-lip_, and appeared at a lower temperature, T_p-am-lip_, than in the first scan. The increase in enthalpy values during the second scan has been related to the better condition for complex formation, because amylose was leaked from the starch granules when temperatures above gelatinisation range were reached in the first scan [43]. In the second scan, no differences in the dissociation enthalpy of the amylose–lipid complex were observed as a consequence of the MWT.

## 4. Conclusions

MWT of quinoa grains proved effective in modifying the morphological, techno-functional, thermal, and rheological properties of the flours. The variation of the treatment conditions (water content and its evolution, constant or variable) strongly influenced the results obtained. The treatments performed at 10% WC led to limited modifications in techno-functional properties, probably due to the lower water availability and the lower temperature reached during the MWT. More intense modifications were generally obtained for higher WC, dependent on the WC evolution during MWT. These differences can be exemplified by the markedly higher pasting profile and crystallinity of sample V-20, in contrast to sample C-30, which showed a lower pasting profile, crystallinity, and resistance of its gels to be broken, probably as a result of its partial gelatinisation. In view of the foregoing, it will be necessary to study the behaviour of the different treated flours in the production of food products to confirm their capacity to improve their technological, nutritional, and sensory quality.

## Figures and Tables

**Figure 1 foods-12-01421-f001:**
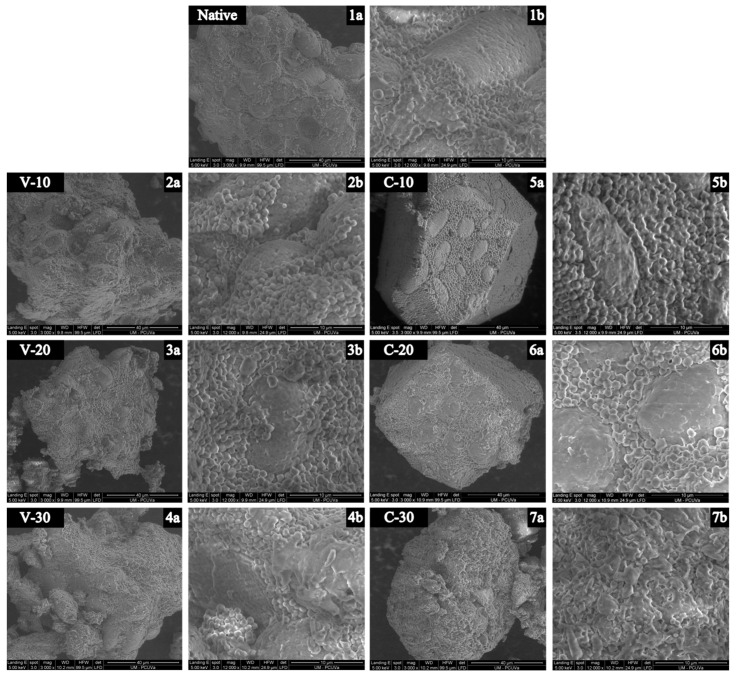
Scanning electron microscopy (SEM) photomicrographs of quinoa flour endosperm (1–7) at different magnifications a: ×3000; b: ×12,000. V-10, V-20, V-30, C-10, C-20, and C-30: samples obtained from untreated (native) and microwaved quinoa grains treated at 10%, 20%, and 30% water content (WC), with variable (V) or constant (C) WC during treatment, respectively.

**Figure 2 foods-12-01421-f002:**
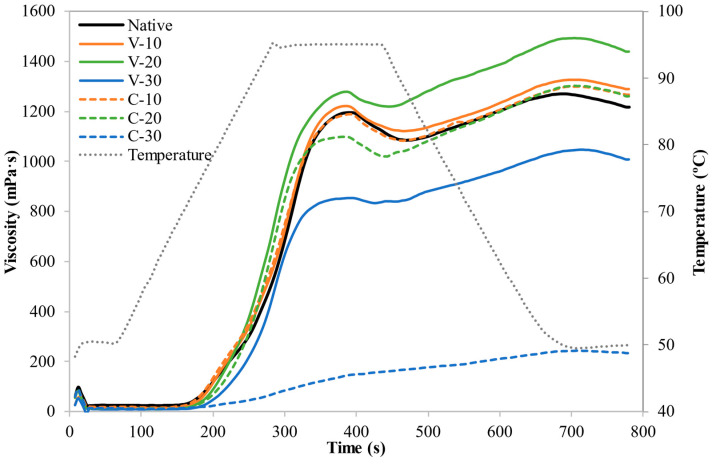
Pasting profiles of untreated flour (Native) and flours obtained from microwaved grains with 10%, 20%, and 30% water content in systems with variable (V) and constant (C) water content. The temperature profile is represented on the second axis.

**Figure 3 foods-12-01421-f003:**
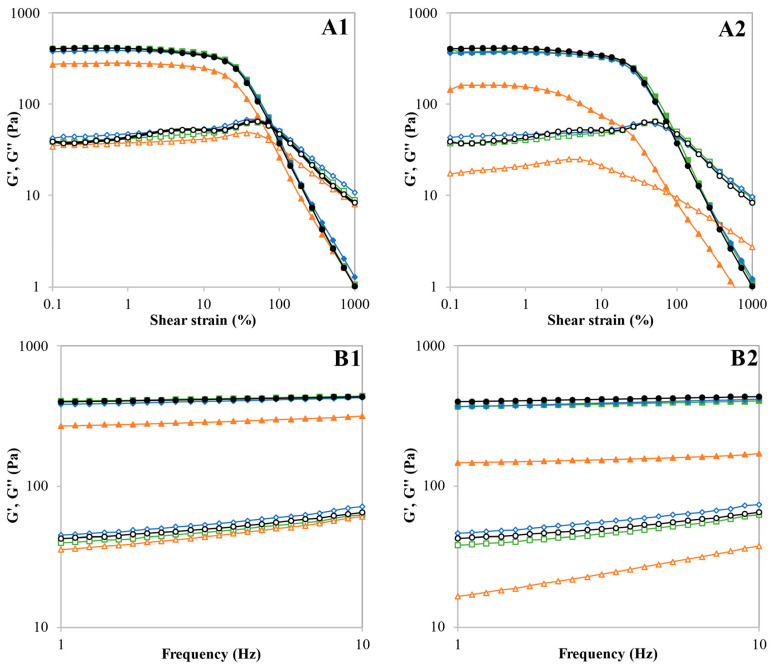
Strain sweeps (**A**) and frequency sweeps (**B**) of the gels made with quinoa flours obtained from untreated (native) and microwaved quinoa grains. Variable water content treatment (V) is presented in the left graphs (**1**) and constant water content treatment (C) in the right graphs (**2**). Native flour is represented in both graphs. Symbol and colour assignment: Native flour (black, ●○), V-10 and C-10 (green, ◼◻), V-20 and C-20 (blue, ◆◇), V-30 and C-30 (orange, ▲△). G′ is represented by full symbols and G′′ by empty symbols.

**Figure 4 foods-12-01421-f004:**
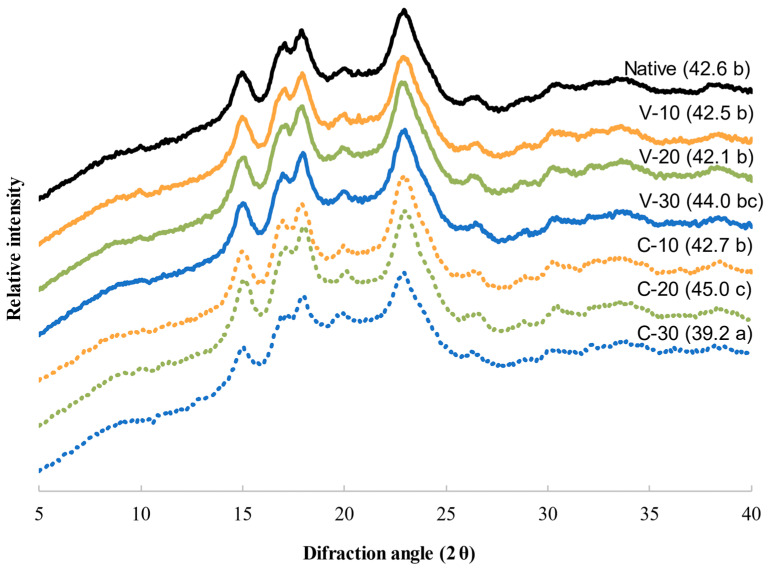
X-ray diffraction patterns (XRD) of untreated flour (native) and flours obtained from microwaved grains with 10, 20, and 30% water content with treatment systems of variable (V) and constant (C) water content. The percentage of crystallinity of each sample is shown in brackets. Mean values with different letters for the same parameter imply significant differences between means at *p* < 0.05.

**Figure 5 foods-12-01421-f005:**
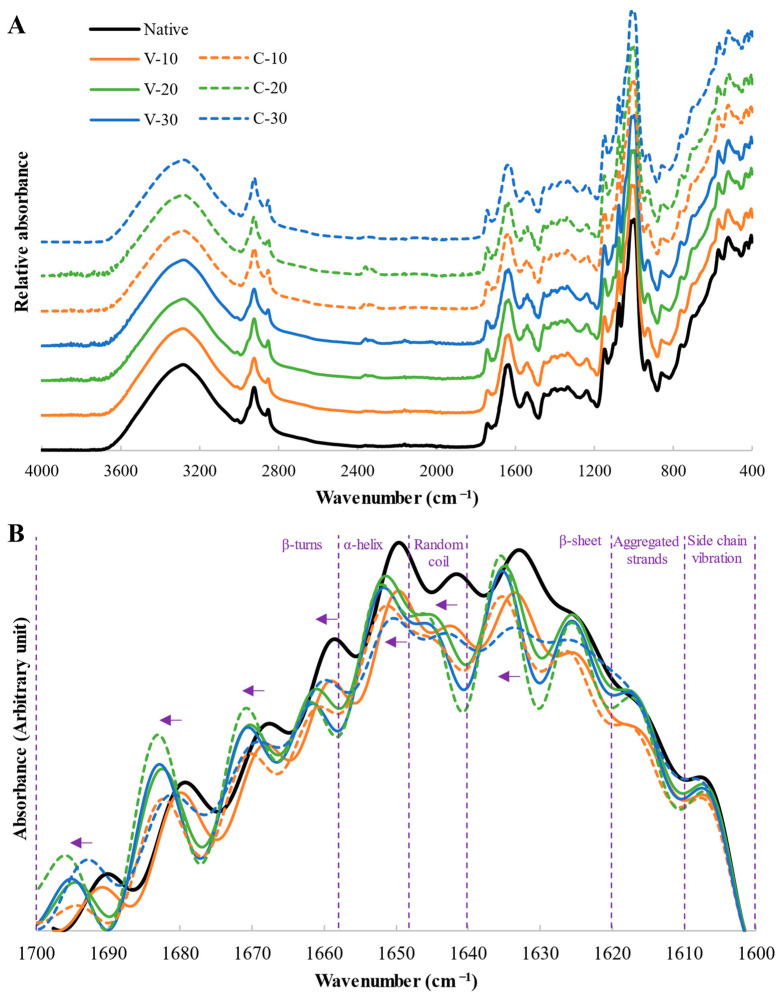
(**A**) FTIR spectra of the untreated flour (native) and flours obtained from microwaved grains with 10, 20, and 30% water content with treatment systems of variable (V) or constant (C) water content. (**B**) Deconvoluted FTIR spectra of the Amide I region. The different structures are represented, and the arrows indicate the wavenumber shift in the treated samples.

**Table 1 foods-12-01421-t001:** Maximum temperature reached and final water content during microwave treatment, colour characteristics and particle size distribution of the flour samples obtained from untreated (native) and microwave-treated quinoa grains.

Sample	Final WC (%)	T_max_ (°C)	Colour Characteristics				Particle Size Distribution		
			L*	C*	h	ΔE	D_10_ (µm)	D_50_ (µm)	D_90_ (µm)
Native	-	-	83.1 bc	12.3 a	72.8 cd	-	17 b	157 e	371 d
V-10	8 a	77 a	83.7 c	13.0 b	73.0 d	1.2 ab	17 b	135 c	328 b
V-20	14 c	82 a	83.5 c	13.0 b	72.6 c	1.0 a	14 a	122 a	314 a
V-30	24 e	88 ab	82.2 b	13.0 b	72.3 b	1.5 b	18 b	128 ab	321 ab
C-10	10 b	83 a	83.2 bc	13.3 b	72.8 cd	1.8 b	20 c	150 d	352 c
C-20	20 d	88 ab	82.9 bc	13.2 b	72.2 b	1.3 ab	18 bc	134 bc	324 b
C-30	30 f	99 b	76.6 a	19.4 c	67.7 a	9.8 c	18 bc	129 bc	327 b
SE	0.5	4	0.4	0.1	0.1	0.2	1	2	3
Analysis of variance and significance (*p*-values)									
WC	**0.015**	**0.033**	**0.000**	**0.000**	**0.000**	**0.000**	**0.004**	**0.000**	**0.000**
System	**0.000**	0.065	**0.000**	**0.000**	**0.000**	**0.000**	**0.000**	**0.000**	**0.000**
WC × System	**0.027**	0.784	**0.000**	**0.000**	**0.000**	**0.000**	**0.027**	**0.004**	0.054

V-10, V-20, V-30, C-10, C-20, and C-30: Samples obtained from microwaved quinoa grains treated at 10%, 20%, and 30% water content (WC), and with treatment systems of variable (V) or constant (C) WC, respectively; T_max_: maximum temperature reached during microwave treatment; L*: luminosity; C*: chroma; h: hue; ΔE: colour difference from native flour; D_10_: diameter where 10% of particles had smaller particle size; D_50_: median diameter, diameter where 50% of particles had smaller particle size; D_90_: diameter where 90% of particles had smaller particle size. Mean values with different letters for the same parameter imply significant differences between means at *p* < 0.05. SE: Pooled standard error obtained from ANOVA. Bolded *p*-values (*p* < 0.05) mean that the effects of WC, system, and their interaction on microwave treatment are significant at >95% confidence level.

**Table 2 foods-12-01421-t002:** Techno-functional properties of flour samples obtained from untreated (native) and microwave-treated quinoa grains.

Sample	EA (%)	ES (%)	FC (mL)	FS (%)	WAC (g/g)	OAC (g/g)	WAI (g/g)	WSI (g/100g)	SP (g/g)
Native	51.2 e	16.8 f	6.0 d	17 c	0.96 a	1.17 b	11.5 c	7.9 c	12.5 d
V-10	51.5 e	14.4 e	5.5 cd	15 c	0.97 a	1.17 b	11.8 d	7.7 bc	12.8 e
V-20	10.5 d	5.0 c	4.5 c	0 a	1.21 b	1.11 a	10.8 ab	7.2 a	11.6 ab
V-30	6.5 c	2.7 a	2.0 b	0 a	1.58 c	1.11 a	10.6 a	7.8 c	11.5 a
C-10	50.8 e	12.6 d	7.5 e	6 b	0.96 a	1.21 b	12.2 e	7.3 ab	13.2 f
C-20	2.4 b	3.9 b	1.8 b	0 a	1.61 d	1.19 b	11.0 b	7.0 a	11.8 bc
C-30	0.0 a	-	0.3 a	0 a	2.04 e	1.08 a	11.0 b	8.5 d	12.1 c
SE	0.3	0.3	0.4	1	0.01	0.02	0.1	0.2	0.1
Analysis of variance and significance (*p*-values)									
WC	**0.000**	**0.000**	**0.000**	**0.000**	**0.000**	**0.000**	**0.000**	**0.000**	**0.000**
System	**0.000**	**0.000**	**0.037**	**0.004**	**0.000**	0.060	**0.001**	0.888	**0.000**
WC × System	**0.000**	**0.030**	**0.002**	**0.002**	**0.000**	**0.026**	0.596	**0.024**	0.191

V-10, V-20, V-30, C-10, C-20, and C-30: Samples obtained from microwaved quinoa grains treated at 10%, 20%, and 30% water content (WC), and with treatment systems of variable (V) or constant (C) WC, respectively; EA: emulsifying activity; ES: emulsion stability: FC: foam capacity; FS: foam stability; WAC: water absorption capacity; OAC: oil absorption capacity; WAI: water absorption index; WSI: water solubility index; SP: swelling power. Mean values with different letters for the same parameter imply significant differences between means at *p* < 0.05. SE: Pooled standard error obtained from ANOVA. Bolded *p*-values (*p* < 0.05) mean that the effects of WC, system, and their interaction on microwave treatment are significant at >95% confidence level.

**Table 3 foods-12-01421-t003:** Pasting properties of flour samples and rheological properties of gels obtained from untreated (native) and microwave-treated quinoa grains.

Sample	PT (°C)	PV (mPa·s)	TV (mPa·s)	BV (mPa·s)	FV (mPa·s)	SV (mPa·s)	G′_1_ (Pa)	a	G″_1_ (Pa)	b	(tan δ)_1_	c	τ_max_ (Pa)	Crosspoint (Pa)
Native	72.5 a	1198 e	1079 d	120 f	1215 c	137 b	400 d	0.039 a	40 c	0.187 a	0.100 a	0.149 a	45 d	63 d
V-10	73.8 b	1224 f	1128 e	96 e	1291 e	164 c	416 d	0.036 a	41 cd	0.192 a	0.099 a	0.156 ab	47 d	67 e
V-20	75.0 c	1279 g	1221 f	58 c	1443 f	222 e	369 c	0.053 b	44 de	0.202 ab	0.118 b	0.149 a	41 c	63 d
V-30	76.6 d	853 b	835 b	18 b	1009 b	174 cd	262 b	0.066 c	35 b	0.239 c	0.133 c	0.174 b	31 b	44 b
C-10	73.1 ab	1176 d	1069 d	107 e	1254 d	185 d	369 c	0.039 a	37 b	0.209 b	0.100 a	0.170 b	43 cd	67 e
C-20	76.7 d	1109 c	1028 c	82 d	1278 de	250 f	369 c	0.054 b	45 e	0.199 ab	0.123 b	0.145 a	44 cd	60 c
C-30	79.4 e	158 a	158 a	0 a	233 a	75 a	156 a	0.057 b	15 a	0.362 d	0.100 a	0.306 c	3 a	11 a
SE	0.3	6	7	4	8	4	5	0.003	1	0.005	0.002	0.006	1	1
Analysis of variance and significance (*p*-values)														
WC	**0.002**	**0.000**	**0.000**	**0.023**	**0.000**	**0.003**	**0.000**	0.447	**0.000**	**0.000**	**0.001**	**0.000**	**0.000**	**0.000**
System	**0.000**	**0.000**	**0.000**	**0.000**	**0.000**	**0.000**	**0.000**	**0.000**	**0.000**	**0.000**	**0.000**	**0.000**	**0.000**	**0.000**
WC × System	**0.002**	**0.000**	**0.000**	**0.000**	**0.000**	**0.000**	**0.000**	0.131	**0.000**	**0.000**	**0.000**	**0.000**	**0.000**	**0.000**

V-10, V-20, V-30, C-10, C-20, and C-30: Samples obtained from microwaved quinoa grains treated at 10%, 20%, and 30% water content (WC), and with treatment systems of variable (V) or constant (C) WC, respectively; PT: pasting temperature; PV: peak viscosity, TV: trough viscosity, BV: breakdown viscosity; FV: final viscosity; SV: setback viscosity. The power law model was fitted to the frequency sweep experimental data (G′ = G′_1_·ω^a^; G″ = G″_1_·ω^b^; tan δ = (tan δ)_1_·ω^c^), where G_1_′, G_1_″, and tan(δ)_1_ are the coefficients obtained from the fitting and represent the elastic, viscous moduli, and loss tangent, respectively, at a frequency of 1 Hz. The a, b, and c exponents quantify the degree of dependence of the dynamic moduli and the loss tangent with the oscillation frequency. τ_max_: Maximum stress that the samples could tolerate in the linear viscoelastic region. Mean values with different letters for the same parameter imply significant differences between means at *p* < 0.05. SE: Pooled standard error obtained from ANOVA. Bolded *p*-values (*p* < 0.05) mean that the effects of WC, system, and their interaction on microwave treatment are significant at >95% confidence level.

**Table 4 foods-12-01421-t004:** Starch bands and protein secondary structure content from FTIR analysis of flour samples obtained from untreated (native) and microwave-treated quinoa grains.

Sample	Starch Bands		Protein Secondary Structure Analysis (%)					
	IR 1047/1022	IR 1022/995	Side Chain Vibration	Aggregated Strands	β-Sheet	Random Coil	α-Helix	β-Turn
Native	0.80 c	0.94 abc	4.0 a	10.5 a	27.7 a	13.5 b	16.3 bc	28.1 cd
V-10	0.77 b	0.95 cd	4.0 a	10.7 ab	29.8 bc	10.8 a	17.3 c	27.5 bc
V-20	0.76 ab	0.95 bcd	4.3 ab	12.0 cd	29.7 bc	11.4 a	15.5 b	27.1 abc
V-30	0.75 a	0.94 ab	4.4 ab	12.5 d	29.3 b	11.5 a	15.8 b	26.5 ab
C-10	0.76 ab	0.96 d	4.5 b	11.8 cd	30.3 c	11.0 a	16.6 bc	25.9 a
C-20	0.75 a	0.93 a	4.0 a	11.9 cd	27.5 a	13.2 b	13.7 a	29.1 d
C-30	0.75 a	0.96 d	4.7 b	11.4 bc	27.6 a	11.4 a	15.6 b	29.4 d
SE	0.01	0.01	0.1	0.3	0.3	0.4	0.4	0.4
Analysis of variance and significance (p-values)								
WC	**0.043**	0.646	**0.034**	**0.044**	**0.000**	**0.008**	**0.001**	**0.018**
System	**0.001**	0.068	0.219	0.801	**0.000**	0.067	**0.025**	**0.010**
WC × System	0.088	**0.009**	**0.032**	**0.008**	**0.001**	0.056	0.187	**0.001**

V-10, V-20, V-30, C-10, C-20, and C-30: Samples obtained from microwaved quinoa grains treated at 10%, 20%, and 30% water content (WC), with treatment systems of variable (V) or constant (C) WC, respectively. Mean values with different letters for the same parameter imply significant differences between means at *p* < 0.05. SE: Pooled standard error obtained from ANOVA. Bolded *p*-values (*p* < 0.05) mean that the effects of WC, system, and their interaction on microwave treatment are significant at >95% confidence level.

**Table 5 foods-12-01421-t005:** Thermal properties of flour samples obtained from untreated (native) and microwave-treated quinoa grains.

Sample	First Scan						Second Scan			
	ΔH_gel_ (J/g dm)	T_p-gel_ (°C)	T_o-gel_ (°C)	T_e-gel_ (°C)	ΔH_am-lip_ (J/g dm)	T_p-am-lip_ (°C)	ΔH_ret_ (J/g dm)	T_p-ret_ (°C)	ΔH_am-lip_ (J/g dm)	T_p-am-lip_ (°C)
Native	7.6 bc	70.7 c	62.3 ab	78.8 a	0.56 c	95.4 c	0.9 ab	48.5 bc	1.5 b	91.7 c
V-10	8.2 d	70.0 b	61.7 a	79.0 ab	0.52 c	94.7 b	0.8 ab	47.1 b	0.7 a	90.9 bc
V-20	7.9 cd	69.5 a	61.6 a	79.0 ab	0.34 b	94.9 bc	0.8 ab	44.3 a	1.0 ab	84.6 a
V-30	7.3 b	70.8 c	63.1 bc	79.4 ab	0.38 b	94.6 b	1.0 b	47.8 bc	1.0 ab	83.7 a
C-10	7.9 cd	69.8 ab	61.8 a	78.9 ab	0.51 c	94.5 b	0.8 ab	43.9 a	1.1 ab	86.3 a
C-20	7.7 bc	69.8 ab	62.2 a	79.1 ab	0.38 b	94.5 b	0.6 a	50.0 c	1.0 ab	85.2 a
C-30	5.3 a	71.4 d	63.6 c	80.2 b	0.22 a	91.2 a	1.4 c	46.2 ab	0.6 a	85.7 ab
SE	0.1	0.1	0.3	0.3	0.02	0.2	0.1	0.8	0.1	1.2
Analysis of variance and significance (*p*-values)										
WC	**0.000**	**0.001**	**0.006**	0.051	**0.004**	**0.000**	**0.009**	0.195	0.625	0.084
System	**0.001**	0.110	0.158	0.305	0.051	**0.000**	0.230	0.724	0.983	0.634
WC × System	**0.002**	0.072	0.630	0.432	**0.016**	**0.001**	0.131	**0.007**	0.240	0.161

V-10, V-20, V-30, C-10, C-20, and C-30: Samples obtained from microwaved quinoa grains treated with 10%, 20%, and 30% water content (WC), and with treatment systems of variable (V) or constant (C) WC, respectively; ΔH_gel_: starch gelatinisation associated enthalpy; T_p-gel_, T_o-gel_, T_e-gel_: peak, onset, and endset temperatures for gelatinisation peak; ΔH_am-lip_: enthalpy associated to dissociation of amylose–lipid complex; T_p-am-lip_: peak temperature for amylose–lipid complex dissociation peak; ΔH_ret_: enthalpy associated to the melting of recrystallized amylopectin; T_p-ret_: peak temperature of melting of recrystallized amylopectin; dm: dry matter. Mean values with different letters for the same parameter imply significant differences between means at *p* < 0.05. SE: Pooled standard error obtained from ANOVA. Bolded *p*-values (*p* < 0.05) mean that the effects of WC, system, and their interaction on microwave treatment are significant at >95% confidence level.

## Data Availability

The data presented in this study are available on request from the corresponding author.

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
