# Peer review of "Microwave Modification of Quinoa Grains at Constant and Varying Water Content Modulates Changes in Structural and Physico-Chemical Properties of the Resulting Flours"

_foods, 2023, doi:10.3390/foods12071421_

Round 1
Reviewer 1 Report
Dear Authors, after reviewing the article “Microwave modification of quinoa grains at constant and varying water content modulates changes in structural and physico-chemical properties of the resulting flours” I suggest reviewing the following points:
Line 104, Concanavalin A is more understandable.
Throughout the article there are errors in the correct symbol to represent the degrees, please correct.
Lines 119-120, describe more clearly the external agitation device, it is not understood how the process is or how the equipment is.
What was the reason for conditioning the samples at the end at 11% humidity and not at another percentage?
Line 159, why the EA and ES methods are described and the rest are not?
Line 219, is the reference well cited?
Table 3, improve the presentation, it looks overloaded.
I must also comment that the article has minimal errors and a good discussion of results. In general, the authors present an interesting article with a remarkable presentation of results.
Author Response
Please, refer to the attached file

Reviewer 2 Report
The authors have made great efforts to explore the feasibility of MWT of quinoa grain under 10%, 20% and 30% of WC to modify the morphological structure and techno-functional, thermal and rheological properties of quinoa flour. The findings are of interest to readers, food scientists and who are working in the food processing industry. However, some revisions and edits should be made to improve the quality of the paper. The authors can find some comments directly highlighted in the pdf file. Some notes are listed below:
1. Why did the authors select WC 10%, 20% and 30% to treat the grains of quinoa
2. Why did the authors select 7d and 4oC? how about the others?And should explain and discuss in the results and discussion part.
3. The section of Statistical analysis must be reworded (see the comments).
4. Results and discussion should be explained more about possible those conditional treatments cause any adverse influence of quinoa quality.
5. Conclusions must be reworded (see comments)

Author Response
Please, refer to the attached file

Reviewer 3 Report
In this work, the authors evaluated the possible application of microwave modification under different conditions to modify the structural and physicochemical properties of quinoa flour. The topic is interesting and all parts of the manuscript are well-written. However, I have two questions:
1. Did the treatments obtained by System C dry after modification? Because normally, after hydrothermal treatment, the flour should be dried to minimize the effect of water in the system.
2. As shown in Table 2, almost all techno-functional properties of flour samples were reduced by applying microwave treatment. So, what is the authors' justification for this phenomenon? Can it be called modification?
Author Response
Please, refer to the attached file

Round 2
Reviewer 2 Report
The manuscript has been significantly revised. All comments and suggestions have been solved.
Reviewer 3 Report
The manuscript is now acceptable.